# Predictive Value of Hepatitis B Core-Related Antigen for Multiple Recurrence Outcomes After Treatment Cessation in Chronic Hepatitis B: A Meta-Analysis Study

**DOI:** 10.3390/v17070929

**Published:** 2025-06-30

**Authors:** Guoyang Yu, Meiqi Cheng, Yuxin Duan, Minrong Kang, Ning Jiang, Wei Yan, Jianhua Yin

**Affiliations:** 1Department of Laboratory Medicine, Naval Medical Centre, Naval Medical University, Shanghai 200052, China; 17621191639@163.com (G.Y.); 13235733612@163.com (M.C.); 13701937303@163.com (Y.D.); abclsab@163.com (M.K.); jiangning990921@163.com (N.J.); 2Department of Epidemiology, Naval Medical University, Shanghai 200433, China

**Keywords:** HBcrAg, meta-analysis, viral relapse prediction, CHB patients, treatment cessation strategy

## Abstract

Background: Hepatitis B core-related antigen (HBcrAg), a novel serum biomarker reflecting the activity of intrahepatic covalently closed circular DNA (cccDNA), has generated conflicting evidence regarding its clinical utility for predicting post-antiviral therapy relapse in chronic hepatitis B (CHB) patients. Methods: We systematically analyzed 13 studies (15 cohorts, *n* = 1529 patients) from PubMed, Web of Science, Wanfang, and CNKI (through April 2025). A bivariate model evaluated HBcrAg’s predictive performance for relapse outcomes, including virological relapse, clinical relapse, and hepatitis flares. Results: HBcrAg demonstrated a pooled sensitivity of 0.81 (95% CI: 0.75–0.86) and specificity of 0.72 (95% CI: 0.67–0.76) for relapse prediction, with a diagnostic odds ratio of 10.66 (95% CI: 7.36–15.42) and summary AUC of 0.83 (95% CI: 0.80–0.86). Subgroup analysis identified threshold effects as the primary source of heterogeneity, which resolved (I^2^ < 13%) after excluding studies with outlier cutoff values. Meta-regression established that HBcrAg’s predictive value was unaffected by age, sex, hepatitis B e antigen status, or detection methods (*p* > 0.05). Conclusions: HBcrAg is validated as a robust non-invasive biomarker to optimize treatment cessation strategies, with high sensitivity providing strong negative predictive value in CHB populations. Future research should prioritize multi-marker models to enhance prediction accuracy.

## 1. Introduction

According to 2022 data, chronic hepatitis B virus (HBV) infection affects approximately 257 million individuals globally, contributing to over 800,000 annual deaths from complications including cirrhosis and hepatocellular carcinoma (HCC) [1]. Significant diagnostic and therapeutic gaps persist: only 13% of chronic hepatitis B (CHB) patients are diagnosed, and fewer than 3% receive antiviral therapy [2,3,4].

A key barrier to cure is the persistence of covalently closed circular DNA (cccDNA) in hepatocytes. Resisting elimination by current antivirals, cccDNA serves as a molecular reservoir that maintains viral persistence for decades—even after apparent infection resolution. This reservoir can reactivate viral replication, triggering disease relapse following therapy cessation [5,6].

Hepatitis B core-related antigen (HBcrAg) has recently emerged as a serum biomarker strongly correlated with intrahepatic cccDNA activity (r = 0.641, 95% confidence interval [CI]: 0.510–0.743, *p* < 0.001), outperforming hepatitis B surface antigen (HBsAg) as a virological indicator [5]. Unlike cccDNA quantification requiring invasive liver biopsy, HBcrAg offers a non-invasive surrogate for intrahepatic viral replication [7]. Growing evidence suggests that is has prognostic value for predicting HCC risk and post-treatment recurrence in CHB patients [8], with high sensitivity even at low viral loads (<2000 IU/mL) [9].

Despite these advances, current research focuses predominantly on single recurrence endpoints. No systematic evaluation exists for HBcrAg’s predictive efficacy across multidimensional outcomes (e.g., virological relapse [VR], clinical relapse [CR], hepatitis flares). Existing meta-analyses [10,11,12] remain limited to traditional biomarkers, overlooking HBcrAg’s unique advantages. This meta-analysis comprehensively evaluates HBcrAg’s predictive capacity for diverse recurrence outcomes, providing evidence to optimize individualized treatment cessation strategies and risk-stratified management in high-risk populations.

## 2. Method

### 2.1. Search Strategy

To evaluate the clinical utility of HBcrAg, we conducted a comprehensive literature search across four databases: PubMed, Web of Science (WOS), Wanfang, and China National Knowledge Infrastructure (CNKI). The specific search strategies were as follows:(1)PubMed: Articles published before 1 April 2025 were retrieved using the search terms “HBcrAg” OR “Hepatitis B core-related Antigen.”(2)WOS: Literature indexed between 1 January 1900 and 1 April 2025 was identified using the keywords “hepatitis B core-related antigen” OR “HBcrAg.”(3)Wanfang: Journal articles, dissertations, and conference papers published from 1 January 1900 to 7 May 2025 were searched for with the terms “hepatitis B core-related antigen,” “HBcrAg,” or “hepatitis B core-related antigen (in Chinese)”.(4)CNKI: All literature from 1 January 1900 to 1 April 2025 was retrieved using the keywords “hepatitis B core-related antigen” OR “HBcrAg” OR “hepatitis B core-related antigen (in Chinese)”.

Following the search, duplicate records were removed, and manual screening was performed to exclude review articles, systematic reviews, practice guidelines, letters, editorials, and non-human studies.

Initial categorization based on study outcomes and designs revealed a substantial number of articles investigating the prognostic value of HBcrAg in post-treatment outcomes in CHB patients. Given the ongoing controversy and divergent findings regarding HBcrAg’s predictive efficacy in clinical outcomes, we focused on studies evaluating HBcrAg as a prognostic biomarker for predicting post-treatment relapse in CHB patients.

### 2.2. Quality Assessment

In this meta-analysis, the Quality in Prognostic Studies (QUIPS) [13] tool, a validated risk of bias assessment instrument, was employed to ensure the methodological rigor of included studies. The updated QUIPS tool categorizes potential biases in prognostic studies into six domains: study participation, study attrition, prognostic factor measurement, outcome measurement, study confounding, and statistical analysis and reporting. Based on descriptions provided in the original studies, the risk of bias within each domain was classified as high, moderate, or low. The QUIPS tool incorporates detailed signaling questions for each dimension, addressing critical aspects such as comprehensive characterization of the study population, recruitment methodology of participants, and reliability of prognostic factor and outcome measurements. By systematically considering these elements, the tool effectively identifies potential biases arising from study design, implementation, and reporting processes, thereby ensuring the scientific validity and reliability of studies included in the analysis.

### 2.3. Data Analysis

Sensitivity, specificity, positive predictive value (PPV), and negative predictive value (NPV) [14] are critical metrics for evaluating the performance of diagnostic tests. In this study, we extracted reported values of PPV, NPV, sensitivity, and specificity from the included articles. Using these metrics alongside the total sample size and the number of positive outcomes, we back-calculated the values of true positives (TP), false positives (FP), true negatives (TN), and false negatives (FN) based on the following formulas. Sensitivity refers to the proportion of true–positive cases correctly identified among all actual positive cases, calculated as TP/(TP + FN). Specificity represents the proportion of true–negative cases correctly identified among all actual negative cases, calculated as TN/(TN + FP). PPV indicates the proportion of true–positive cases among all samples classified as positive, calculated as TP/(TP + FP). NPV denotes the proportion of true–negative cases among all samples classified as negative, calculated as TN/(TN + FN).

The positive likelihood ratio (PLR) and negative likelihood ratio (NLR) were employed to further assess the diagnostic performance. A higher PLR value signifies a stronger ability of the test to differentiate between diseased and non-diseased populations, whereas a lower NLR value suggests superior performance in excluding the disease.

The diagnostic odds ratio (DOR), calculated as the ratio of true positives multiplied by true negatives to false positives multiplied by false negatives (DOR = [TP × TN]/[FP × FN]), integrates information from both sensitivity and specificity to holistically reflect the accuracy of a diagnostic test. A higher DOR value indicates superior diagnostic performance, underscoring its critical role in evaluating the efficacy of diagnostic methodologies. This metric serves as a robust indicator for quantifying test reliability and discriminatory power in clinical decision-making contexts.

The bivariate model was employed to synthesize effect sizes in atypical meta-analyses [15]. This approach enables simultaneous analysis of sensitivity and specificity while accounting for their intrinsic correlation through joint modeling of their distributions. The model generates a summary receiver operating characteristic (SROC) curve, which graphically plots sensitivity on the *y*-axis against 1-specificity on the *x*-axis. This visualization effectively illustrates the distribution of sensitivity–specificity pairs across studies, offering a comprehensive and methodologically rigorous evaluation of diagnostic test performance by integrating multidimensional accuracy metrics into a unified analytical framework.

Furthermore, threshold effects were examined using Spearman’s correlation analysis between sensitivity and 1-specificity. Statistical significance was determined by a *p*-value threshold of <0.05, with values below this threshold indicating the presence of significant threshold effects. Concurrently, heterogeneity attributable to covariates beyond threshold effects was quantified through the I^2^ statistic, which estimates the proportion of total variation across studies that reflects genuine differences rather than random chance. The I^2^ index was interpreted using established thresholds: values < 25% denote low heterogeneity, 25–50% suggest moderate heterogeneity, and >50% indicate substantial heterogeneity. This dual analytical approach enables systematic differentiation between threshold-induced variability and other sources of study-level heterogeneity, thereby enhancing the interpretative validity of pooled diagnostic accuracy estimates.

Subgroup analysis stratifies study populations according to predefined variables and conducts independent meta-analyses within each stratum, with the objective of investigating between-subgroup heterogeneity in effect magnitudes and identifying potential sources of variability.

Meta-regression extends this approach by modeling study-level effect sizes as dependent variables and incorporating covariates of interest as independent predictors. This regression-based framework quantitatively evaluates the association between explanatory variables and effect estimates, enabling precise quantification of covariate effects on observed heterogeneity. Both methodologies synergistically address the decomposition of variance components in pooled effect estimates, distinguishing clinically meaningful effect modifiers from stochastic variation.

All statistical analyses were conducted using STATA software (version 17.0; StataCorp., College Station, TX, USA) with the MIDAS module for meta-analytical modeling and Review Manager (version 5.3; The Cochrane Collaboration) for systematic review management. This dual-platform approach ensured methodological rigor in both quantitative synthesis (STATA) and evidence organization (RevMan), aligning with best practices for diagnostic accuracy meta-analyses.

## 3. Results

### 3.1. Study Selection and Characteristics

Initial database searches identified 1555 potentially relevant records. Following deduplication, 940 unique records underwent manual screening. Exclusion criteria encompassed systematic reviews/meta-analyses, mechanistic investigations (e.g., animal/cell studies, non-patient research), case reports, editorial commentaries, conference guidelines, errata, and other non-qualifying publication types. Subsequent full-text review of 280 articles excluded studies focusing on non-target clinical endpoints (e.g., HCC, hepatic fibrosis, cirrhosis, hepatitis staging) or those with inaccessible outcome data (e.g., missing effect estimates, undocumented treatment histories, or non-predictive analyses). Ultimately, 13 articles met the inclusion criteria for quantitative synthesis.

The study selection process based on PRISMA flow reporting standards is detailed in Figure 1, with study characteristics presented in Table 1. The inclusion and exclusion criteria for the literature are detailed in Appendix A.

### 3.2. Risk of Bias Analysis

Risk of bias assessment for the 13 included articles was conducted using the QUIPS tool. Among the studies evaluating HBcrAg’s predictive capacity for post-treatment relapse in CHB patients, six demonstrated low risk of bias, six moderate risk, and one high risk. Detailed results are presented in Table 2.

### 3.3. Predictive Efficacy of HBcrAg Across Various Recurrence Outcomes

This meta-analysis incorporated data from 15 cohorts, utilizing effect sizes corresponding to maximal area under the curve (AUC) values reported in individual studies. For predicting all recurrence outcomes, HBcrAg demonstrated a pooled sensitivity of 0.84 (95% CI, 0.77–0.88; I^2^ = 45.8%) and specificity of 0.67 (95% CI, 0.59–0.74; I^2^ = 87.5%). Diagnostic likelihood ratios included a PLR of 2.56 (95% CI, 2.10–3.11; I^2^ = 74.6%) and NLR of 0.24 (95% CI, 0.18–0.32; I^2^ = 0%), with DOR of 10.53 (95% CI, 7.62–14.57; I^2^ = 33.5%). Bivariate modeling revealed substantial heterogeneity (overall I^2^ = 96.44; 95% CI, 93.81–99.06), predominantly driven by specificity heterogeneity (I^2^ = 87.5%) (Appendix A). Deeks’ funnel plot analysis indicated no significant publication bias (*p* > 0.05) (Figure 2).

Significant threshold effects were identified via Spearman’s correlation analysis of logit-transformed sensitivity and specificity (ρ = −0.6381, *p* < 0.05). Subgroup analysis stratified by cutoff value ranges showed significant heterogeneity in the 3–4 log U/mL group (I^2^ = 87.6%) and the N/A group (I^2^ = 96.3%) (Figure 3). Within the 3–4 log U/mL subgroup, studies by Kaewdech et al. and Fan et al. were outliers, showing results contradicting the pooled estimate with significantly non-overlapping 95% CIs. After exclusion of the N/A group and these two outlier studies, threshold effects were eliminated (ρ = −0.1758, *p* = 0.6218), and heterogeneity was reduced. Pooled analysis demonstrated that HBcrAg achieved a sensitivity of 0.81 (95% CI, 0.75–0.86; I^2^ = 0), specificity of 0.72 (95% CI, 0.67–0.76; I^2^ = 12.61), PLR of 2.8 (95% CI, 2.4–3.3; I^2^ = 0), NLR of 0.27 (95% CI, 0.20–0.35; I^2^ = 0), and DOR of 10.66 (95% CI, 7.36–15.42; I^2^ = 40.99) for predicting recurrence outcomes. The summary AUC was 0.83 (95% CI: 0.80–0.86) (Figure 4 and Figure 5).

Meta-regression analysis of potential confounders (including age, sex, HBeAg positivity rate, HBsAg/HBcrAg levels, publication year, detection timing, and endpoint events) showed no significant impact on HBcrAg’s predictive performance (Appendix A), suggesting robust diagnostic utility across diverse populations and methodological contexts. Furthermore, sensitivity analysis confirmed result stability with <10% variation in the DOR upon sequential study exclusion (Figure 6).

## 4. Discussion

Although NAs effectively suppress HBV replication and delay disease progression, most patients require long-term or lifelong therapy. This challenge stems from the persistence of cccDNA in hepatocytes, which serves as a molecular reservoir for viral reactivation. Even patients achieving serological HBV DNA clearance may experience virological rebound and hepatitis recurrence after treatment cessation. Recently, HBcrAg, a novel biomarker reflecting cccDNA activity and viral reservoir dynamics, has gained significant attention. Studies demonstrate strong correlations between HBcrAg and cccDNA (r = 0.70), intrahepatic total HBV DNA (r = 0.67), and serum HBV DNA (r = 0.69) [29]. Furthermore, declining serum HBcrAg levels correlate with increasing HBsAg seroclearance rates over time [30], and HBcrAg reliably reflects persistent viral reservoir activity during long-term NAs therapy. These properties support its theoretical role in predicting post-treatment relapse.

This meta-analysis systematically evaluated HBcrAg’s predictive performance for multidimensional HBV recurrence outcomes (VR, CR, acute hepatitis flares, and HBV reactivation [HBVr]) using data from 13 studies involving 1529 patients. Results revealed excellent predictive value for endpoints including ALT flares, CR, VR, and HBVr, with a pooled AUC of 0.83 (81% sensitivity and 72% specificity). Meta-regression analysis indicated robust clinical generalizability: covariates including age, sex, HBeAg status, HBsAg levels, and study design variations did not significantly affect predictive stability (interaction tests, *p* > 0.05). Thus, in NA-treated patients meeting current cessation criteria, elevated baseline or EOT serum HBcrAg levels (minimum cutoff: 2.54 log IU/mL in this analysis) may significantly increase relapse risk. This aligns with the Japan Society of Hepatology (JSH) risk stratification strategy, which categorizes HBcrAg levels into three intervals (<3.0 log U/mL, 3.0–4.0 log U/mL, and ≥4.0 log U/mL), assigning relapse risk scores of 0, 1, and 2, respectively. Patients with total scores ≥ 1 exhibit < 50% relapse-free rates and require individualized extended therapy [31].

Current HBV relapse prediction research emphasizes HBsAg, HBV RNA, and HBV DNA. One study found HBsAg significantly inferior to HBcrAg in predicting ALT flares (AUC difference > 0.15, *p* < 0.01) [32]. Although HBV RNA shows comparable predictive potential [32,33], its reliance on complex nucleic acid amplification techniques (e.g., RT-qPCR) limits utility in primary care. Conversely, HBcrAg detection via routine ELISA or CLEIA offers advantages in simplicity, cost-effectiveness, and standardization, enhancing clinical monitoring feasibility.

This study has several limitations. First, the included studies had relatively high bias risk (six moderate-risk, one high-risk), potentially compromising internal validity. Second, while EOT HBcrAg levels are critical for cessation decisions, insufficient data precluded dedicated analysis of this pivotal timepoint. Third, mild specificity heterogeneity persisted even after excluding four studies with aberrant thresholds; potential sources (e.g., detection methods or viral genotype distribution) remain underexplored. Finally, despite robust statistical predictive performance, HBcrAg alone remains insufficient for clinical decisions. JSH guidelines emphasize comprehensive evaluation including HBV DNA, HBsAg quantification, and histological indicators—though this study did not systematically explore multi-marker models.

## 5. Conclusions and Perspectives

Although HBcrAg demonstrates robust predictive utility, its standalone application faces inherent limitations. Future research should prioritize multicenter prospective studies employing standardized HBcrAg assays with predefined thresholds. Such studies incorporate diverse patient cohorts spanning viral genotypes, ethnicities, and liver disease stages to improve generalizability. Furthermore, investigating combinatorial strategies integrating HBcrAg with other virological markers (e.g., HBV RNA and intrahepatic cccDNA) to develop multidimensional predictive models is warranted. This integrated approach will establish a refined risk stratification framework essential for guiding evidence-based antiviral treatment cessation decisions.

## Figures and Tables

**Figure 1 viruses-17-00929-f001:**
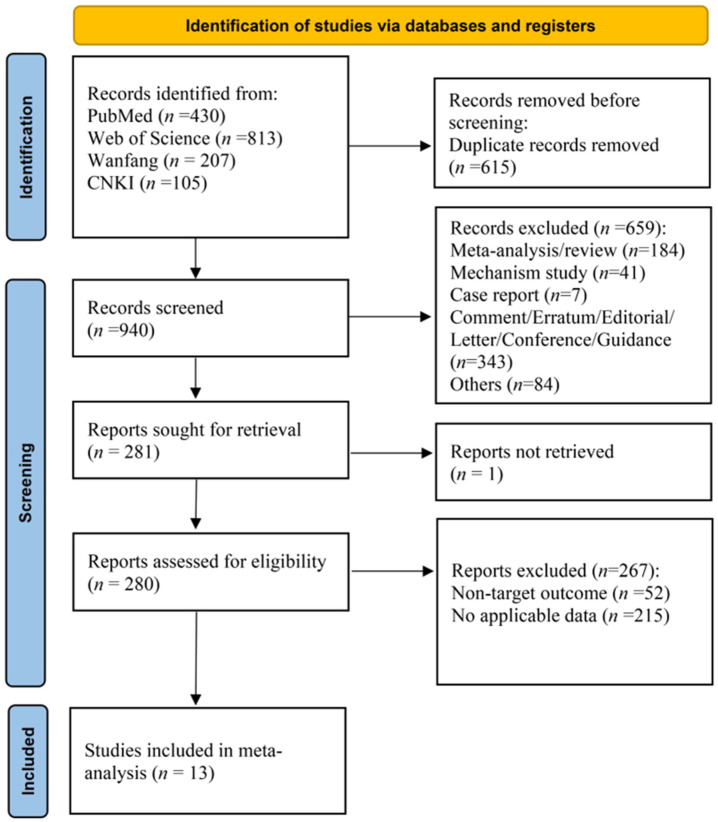
PRISMA flowchart of the study selection process.

**Figure 2 viruses-17-00929-f002:**
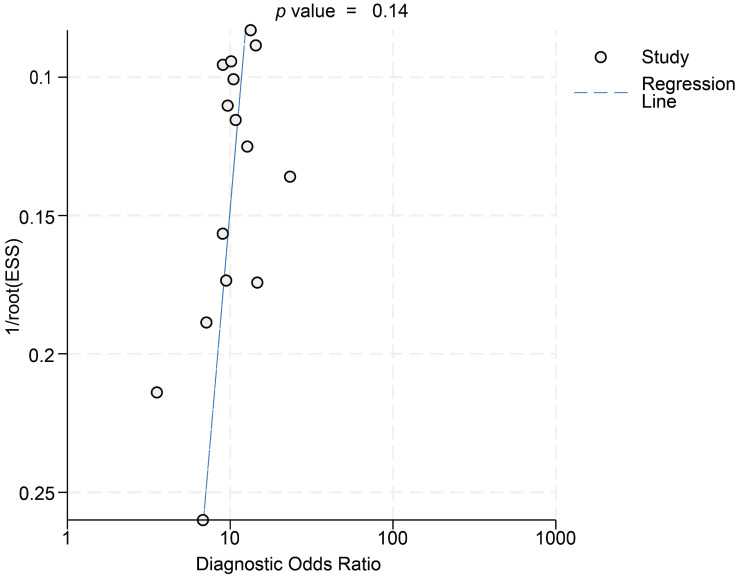
Deeks’ funnel plot asymmetry test.

**Figure 3 viruses-17-00929-f003:**
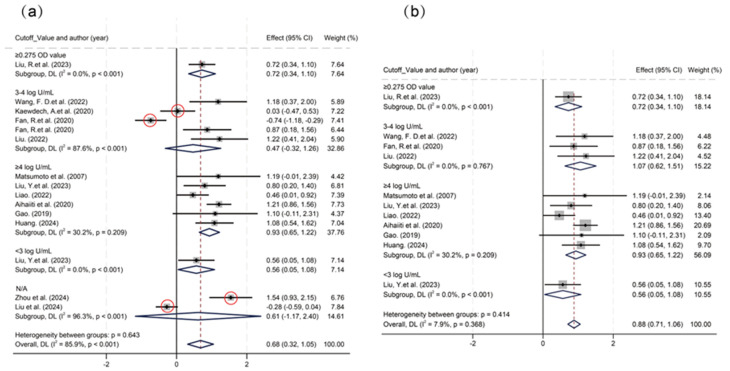
Subgroup analysis based on different cutoff value levels. (**a**) Subgroup analysis based on different cutoff value levels before exclusion. (**b**) Subgroup analysis after exclusion of outlier studies [16,17,18,19,20,21,22,23,24,25,26,27,28].

**Figure 4 viruses-17-00929-f004:**
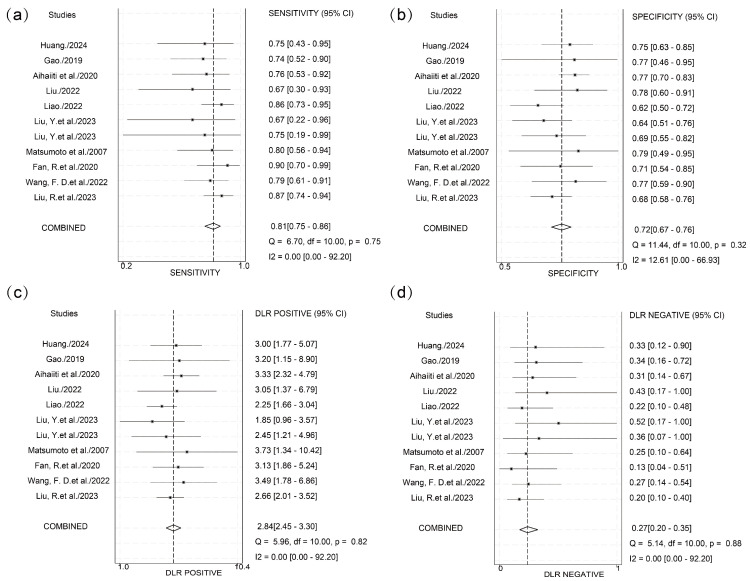
Integrated sensitivity, specificity, PLR, NLR, and DOR after exclusion. (**a**) Forest plot depicting the pooled sensitivity estimate and individual study contributions. (**b**) Forest plot depicting the pooled specificity estimate and individual study contributions. (**c**) Forest plot depicting the pooled PLR estimate and individual study contributions. (**d**) Forest plot depicting the pooled NLR estimate and individual study contributions [17,18,19,20,21,22,25,26,27,28].

**Figure 5 viruses-17-00929-f005:**
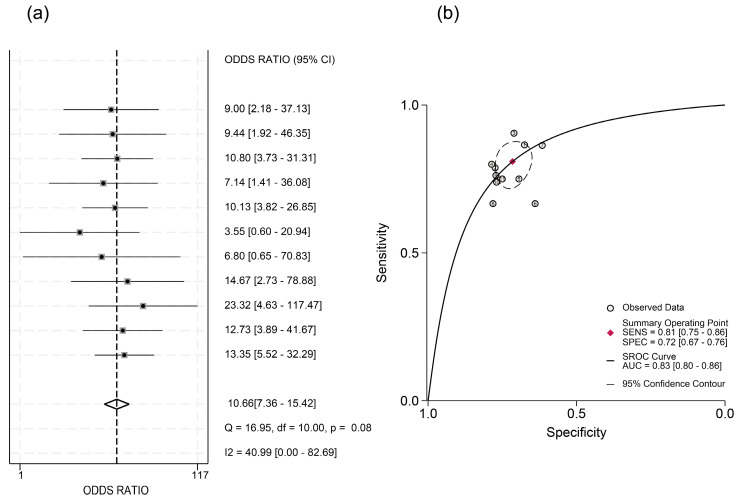
DOR and SROC after exclusion of studies. (**a**) Forest plot depicting the pooled DOR estimate and individual study contributions after exclusion. (**b**) SROC curve post-exclusion with fitted curve, summary operating point (red lozenge), and 95% confidence region (ellipsoid shading) [17,18,19,20,21,22,25,26,27,28].

**Figure 6 viruses-17-00929-f006:**
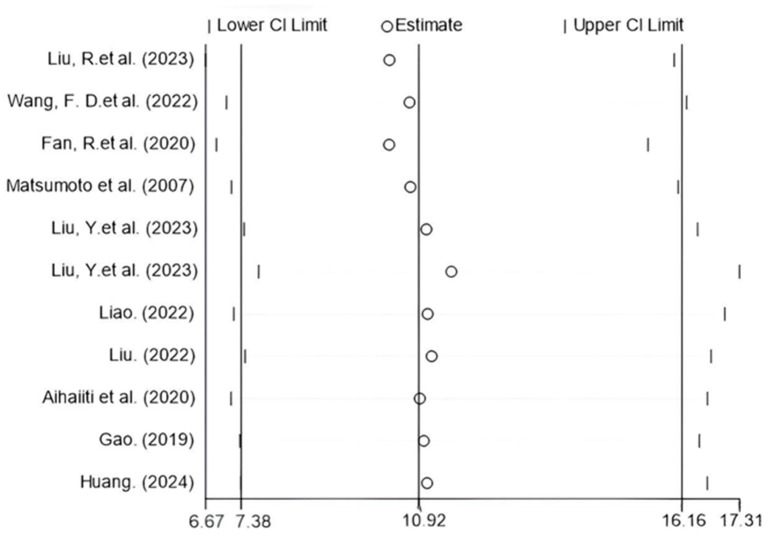
Sensitivity analysis with effect size measured by DOR [17,18,19,20,21,22,25,26,27,28].

**Table 1 viruses-17-00929-t001:** Characteristics of included studies.

Author and Year	Country	Study Type	Sample Year	No. of Patients (F/M)	Baseline HBeAg Status (+/Total Patients)	Baseline HBsAg (Log10 U/mL)	Baseline HBcrAg (Log10 U/mL)	Average Age (Years)	Detection Time/Prediction Time/Outcome	Cut-off Value (Log10 U/mL)	AUC (95% CI) [Sen/Sep]	Method of Measurement
Kaewdech, Apichat et al. (2020) [16]	Thailand	A prospective, observational study	February 2018–August 2019	33/59	20/92	2.96 (2.40–3.43)	3.20 (<3.00–3.90)	55.0 (50.0–63.0)	* EOT/EOT 48 w/Relapse	3	0.773 (0.677–0.869) [90.3%/50.8%]	CLEIA
Liu, Ruyu et al. (2023) [17]	China	A prospective study	January 2019–April 2022	172/0	172/172	>3	<3	26.49 ± 3.77	Postpartum 12-week (12 w) EOT/Postpartum 12 w/Hepatitis B acute flare	0.275 (OD value)	0.84 (0.78–0.91) [86.5%/67.5%]	ELISA
Wang, Fa-Da et al. (2022) [18]	China	A retrospective study	June 2020–January 2021	32/32	36/64	HBeAg(+): 3.41 (2.97–2.47) HBeAg(−): 3.05 (2.77–3.42)	HBeAg(+): 3.34 ± 0.10 HBeAg(−): 3.08 ± 0.09	HBeAg(+): 47.28 ± 6.86 HBeAg(−): 47.39 ± 6.45	EOT/Within 1 year after treatment cessation/Relapse	3.3	0.817 [78%/77%]	CLEIA
Matsumoto, Akihiro et al. (2007) [19]	Japan	A retrospective study	[N/A]	14/20	16/34	positive	6.3 (5.0–7.7)	47 (35–59)	EOT/Within 1 Year After Treatment Cessation/Acute Hepatitis B Flare	4.1–4.6	0.764 [80%/80%]	CLEIA
Fan, Rong et al. (2020) [20]	China	A prospective study	[N/A]	35/92	127/127	3.3 (2.9–3.7)	4.3 (4.0–4.7)	30 (25–35)	EOT/Within 4 Years After EOT/Relapse	4	0.621 [N/A]	CLEIA
				13/46	59/59	2.6 (1.4–3.0)	3.9 (3.3–4.5)	36 (30–43)	EOT/Within 4 Years After EOT/Relapse	4	0.798 [N/A]	CLEIA
Liu, Yiqi et al. (2023) [21]	China	A retrospective study	June 2010–December 2021	23/30	53	Positive	4.27 ± 1.99	53.2 ± 12.0	Pre-chemotherapy/48 Weeks Post-chemotherapy/HBV Reactivation	4.668	0.689 [N/A]	CLEIA
				29/38	67	Negative	2.67 ± 0.54	62.5 ± 11.5	Pre-chemotherapy/48 Weeks Post-chemotherapy/HBV Reactivation	2.54	0.605 [N/A]	CLEIA
Huang, Da et al. (2024) [22]	China	RCT	N/A	14/66	33/80	2.797 ± 0.425	N/A	35.4 ± 9.8	NAs EOT/During Interferon Conversion Therapy/HBV Reactivation	5	0.755 [N/A]	CLEIA
									Baseline HBcrAg Prediction/During Interferon Transition Therapy/HBV Reactivation	N/A	0.713 [N/A]	
Zhou, Fang et al. (2024) [23]	China	A prospective study	June 2020–June 2021	Relapse group: 16/22 Non-relapse group: 36/34	Not mentioned/ 108	2.82 ± 0.08	6.075 ± 1.52	42.83 ± 9.63	EOT/1 Year Later/Post-Treatment Relapse	N/A	0.783 (0.650–0.884) [68.42%/82.68%]	CLEIA
Liu, Yaodan et al. (2024) [24]	China	A prospective observational study	January 2020–October 2022	198/0	198/198	Postpartum Hepatitis Non-flare Group: 3.66 (3.20–4.01) Postpartum Hepatitis Flare Group: 4.05 (3.70–4.24)	Postpartum Hepatitis Non-flare Group: 6.98 (6.77–7.18) Postpartum Hepatitis Flare Group: 7.35 (7.31–7.52)	Postpartum Hepatitis Non-flare Group: 30.89 ± 6.13 Postpartum Hepatitis Flare Group: 29.88 ± 4.55	Antiviral Treatment Initiation/48 Weeks Postpartum/ alt flare	N/A	0.713 [N/A]	ELISA
Liao, Guichan. (2022) [25]	China	A prospective study	November 2011–December 2018	27/95	122/122	2.55 ± 1.11	3.80 ± 0.83	34 (29–40)	EOT/5 Years Post-EOT/Relapse	4.0	0.71 (0.62–0.81) [87.1%/61.5%]	CLEIA
Liu, Zhongwei. (2022) [26]	China	A prospective study	[N/A]	[N/A]	Not mentioned/ 41	2.07 (1.315–2.808)	3.58 ± 0.75	42.09 ± 8.13	EOT/During Follow-up/HBsAg Recurrence	3.8	0.793 [66.67%/78.12%]	CLEIA
Aihaiti et al. (2020) [27]	China	A prospective study	August 2014–June 2017	30/166	Not mentioned/ 196	N/A	4.9 (2–7.5)	55 (40–74)	Post-LT Day 3/3 Years Post-Transplant/HBV Recurrence	4.5	0.818 (0.72–0.917) [76.19%/77.14%]	CLEIA
Gao, Chang. (2019) [28]	China	A prospective study	January 2003–July 2011	3/33	25/36	2.78 (2.02–3.32)	3.85 (3.0–4.4)	35.5 (29.25–41)	EOT 12-week/EOT 48-week/Relapse	4.1	0.719 (0.536–0.902) [73.7%/76.9%]	CLEIA

* EOT: end-of-treatment; w: week(s); N/A: not available; ELISA: enzyme-linked immunosorbent assay; CLEIA: chemiluminescent immunoassay.

**Table 2 viruses-17-00929-t002:** Risk of bias assessment of included studies.

Author (Year)	Study Participation	Study Attrition	Prognostic Factor Measurement	Outcome Measurement	Study Confounding	Statistical Analysis and Reporting	Overall Risk of Bias
Kaewdech, Apichat et al. (2020) [16]	Low	Low	Low	Low	Low	Low	Low
Liu, Ruyu et al. (2023) [17]	Low	Low	Low	Low	Low	Low	Low
Wang, Fa-Da et al. (2022) [18]	Low	Low	Low	Low	Low	Low	Low
Matsumoto, Akihiro et al. (2007) [19]	Moderate	Low	Low	Low	Low	Low	Moderate
Fan, Rong et al. (2020) [20]	Low	Moderate	Low	Low	Low	Low	Moderate
Liu, Yiqi et al. (2023) [21]	Moderate	Low	Low	Low	Low	Low	Moderate
Huang, Da et al. (2024) [22]	Low	Low	Low	Low	Low	Low	Low
Zhou, Fang et al. (2024) [23]	Low	Low	Moderate	Low	Low	Low	Moderate
Liu, Yaodan et al. (2024) [24]	Low	Moderate	High	Low	Low	Low	High
Liao, Guichan. (2022) [25]	Low	Low	Low	Low	Low	Low	Low
Liu, Zhongwei. (2022) [26]	Moderate	Low	Low	Low	Low	Low	Moderate
Aihaiti et al. (2020) [27]	Low	Low	Low	Low	Low	Low	Low
Gao, Chang. (2019) [28]	Moderate	Low	Low	Low	Low	Low	Moderate

## Data Availability

The datasets used and/or analyzed during the current study are available from the corresponding authors on reasonable request. Please contact yinjh@smmu.edu.cn for data access inquiries.

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
