# Peer review of "Predictive Value of Hepatitis B Core-Related Antigen for Multiple Recurrence Outcomes After Treatment Cessation in Chronic Hepatitis B: A Meta-Analysis Study"

_viruses, 2025, doi:10.3390/v17070929_

Round 1

Reviewer 1 Report

Comments and Suggestions for Authors

This is an interesting meta-analysis on HBcrAg as a predictive marker of the outcomes of treatment discontinuation. 

Comments

1. The title (page 1, line 3) should include “after treatment discontinuation” to reflect the contents of the manuscript

2. Page 1, line 38 to line 43 This section should be removed from the manuscript. It is irrelevant.

3. Table 1: Please add a footnote explaining abbreviations (“EO” is EOT?)

4. Page 2, line 87: It would be clinically significant to include only studies with HBcrAg measurements at EOT as this point is the important one for making the decision for treatment discontinuation. If not possible, at least add this point at page 14, line 274, the section of the limitations of this study.

Author Response

Comment 1: The title (page 1, line 3) should include “after treatment discontinuation" to reflect the contents of the manuscript.

Response: Thank you for this comment. According to this comment, the paper title has been revised to Predictive Value of Hepatitis B Core-Related Antigen for Multiple Recurrence Outcomes after Treatment Cessation in Chronic Hepatitis B: A Meta-Analysis Study

Comment 2: Page 1, line 38 to line 43 This section should be removed from the manuscript. lt is irrelevant.

Response: Thank you for this comment. The section spanning Page 1, lines 38–43 has been removed from the manuscript as suggested.

Comment 3: Table 1: Please add a footnote explaining abbreviations (“EO" is EOT?)

Response: Thank you for this comment. A footnote has been added to Table 1 to clarify all abbreviations. The notation “EO” has been corrected to “EOT” (End of Treatment) where applicable.

Comment 4: Page 2, line 87: lt would be clinically significant to include only studies with HBcrAg measurements at EOT as this point is the important one for making the decision for treatment discontinuation. lf not possible, at least add this point at page 14, line 274, the section of the limitations of this study.

Response: Thank you for this professional comment. We acknowledge the importance of HBcrAg measurements at the End of Treatment (EOT) for clinical decision-making. However, a subset of included studies did not explicitly define HBcrAg measurement timing as EOT. This limitation has been explicitly detailed in the Limitations section of the Discussion.

Reviewer 2 Report

Comments and Suggestions for Authors
  1. In Figure 1,one of the screening criteria is “Times Noew Roman”, what do the authors mean?
  2. In Table 2, why the author highlighted the Author (year) of the first study by using yellow color?
  3. In Figure 3, four studies were excluded to re-evaluated the heterogeneity, however, the criteria used to excluded these four studies should be more clearly described.
  4. The font size of Figure 4 and Figure 5 are too small.

Author Response

Comment 1: In Figure 1, one of the screening criteria is “Times Noew Roman", what do the authors mean?

Response 1: Thank you for this thoughtful comment and time spent on reviewing this project. The text "Times Noew Roman" was a typographical error in the original figure. This has been corrected in the revised version of Figure 1 to accurately reflect the intended screening criteria.

Comment 2: In Table 2, why the author highlighted the Author (year) of the first study by using yellow color?

Response 2: Thank you for this comment. The yellow highlighting was unintentional and has been removed from Table 2 to ensure consistent formatting across all entries.

Comment 3: In Figure 3, four studies were excluded to re-evaluated the heterogeneity, however, the criteria used to excluded these four studies should be more clearly described.

Response 3: Thank you for highlighting this. The exclusion criteria for the four studies have been explicitly defined in the revised manuscript (lines 269–280): 1) Two outliers (Kaewdech et al. and Fan et al.) in the 3–4 log U/mL group showed discordant effects (opposite directionality and non-overlapping 95% CIs); 2) The entire "N/A" cutoff subgroup (2 studies) exhibited extreme heterogeneity and lacked clinically interpretable thresholds.

Comment 4: The font size of Figure 4 and Figure 5 are too small.

Response 4: Thank you for this comment. The font sizes in Figures 4 and 5 have been uniformly increased to enhance readability while maintaining visual coherence.